# Investigating the Effects of a Focal Muscle Vibration Protocol on Sensorimotor Integration in Healthy Subjects

**DOI:** 10.3390/brainsci13040664

**Published:** 2023-04-15

**Authors:** Nicoletta Manzo, Francesca Ginatempo, Daniele Belvisi, Giorgio Arcara, Ilaria Parrotta, Giorgio Leodori, Franca Deriu, Claudia Celletti, Filippo Camerota, Antonella Conte

**Affiliations:** 1IRCCS San Camillo Hospital, Via Alberoni 70, 30126 Venice, Italy; 2Department of Biomedical Sciences, University of Sassari, Viale San Pietro 43c, 07100 Sassari, Italy; 3Department of Human Neurosciences, Sapienza University of Rome, Viale dell’Università 30, 00185 Rome, Italy; 4IRCCS Neuromed, Via Atinense 18, 86077 Pozzilli, Italy; 5Movement Contral and Neuroplasticity Research Group, Tervuursevest 101, 3001 Leuven, Belgium; 6Unit of Endocrinology, Nutritional and Metabolic Disorders, AOU Sassari, 07100 Sassari, Italy; 7Physical Medicine and Rehabilitation Division, Umberto I University Hospital of Rome, 00185 Rome, Italy

**Keywords:** focal muscle vibration, sensorimotor integration, neuromodulation

## Abstract

**Background**: The ability to perceive two tactile stimuli as asynchronous can be measured using the somatosensory temporal discrimination threshold (STDT). In healthy humans, the execution of a voluntary movement determines an increase in STDT values, while the integration of STDT and movement execution is abnormal in patients with basal ganglia disorders. Sensorimotor integration can be modulated using focal muscle vibration (fMV), a neurophysiological approach that selectively activates proprioceptive afferents from the vibrated muscle. **Method**: In this study, we investigated whether fMV was able to modulate STDT or STDT–movement integration in healthy subjects by measuring them before, during and after fMV applied over the first dorsalis interosseous, abductor pollicis brevis and flexor radialis carpi muscles. **Results**: The results showed that fMV modulated STDT–movement integration only when applied over the first dorsalis interosseous, namely, the muscle performing the motor task involved in STDT–movement integration. These changes occurred during and up to 10 min after fMV. Differently, fMV did not influence STDT at rest. We suggest that that fMV interferes with the STDT–movement task processing, possibly disrupting the physiological processing of sensory information. **Conclusions**: This study showed that FMV is able to modulate STDT–movement integration when applied over the muscle involved in the motor task. This result provides further information on the mechanisms underlying fMV, and has potential future implications in basal ganglia disorders characterized by altered sensorimotor integration.

## 1. Introduction

The ability to temporally discriminate tactile stimuli can be measured with the somatosensory temporal discrimination threshold (STDT) technique, defined as the shortest time interval necessary for two sequential tactile stimuli to be perceived as consecutive [1,2,3]. STDT involves the activation of cortical structures, specifically the inhibitory interneuronal circuits in the primary somatosensory cortex (S1) [4,5] and the subcortical structures, i.e., the basal ganglia and thalamus [1,6]. Previous studies in healthy humans have shown that STDT values are modulated by voluntary movement execution [7,8]. Movement execution increases STDT values through mechanisms of sensorimotor integration, modulating the physiological gating of tactile inputs that have been hypothesized to occur in basal ganglia–thalamus connections in healthy humans [7,9]. Supporting this hypothesis, studies in patients with movement disorders, which mainly underlie basal ganglia abnormalities as well as dysfunctions in sensorimotor integration, have shown abnormal STDT–movement integration compared to healthy controls [10,11,12,13].

A neurophysiological approach that may potentially modulate sensorimotor integration is muscle vibration [14,15,16,17]. Mechanical vibration of a human limb, applied on a muscle tendon or the belly, allows proprioceptive afferents, namely, group Ia muscle spindle afferents, to be preferentially stimulated [18]. Previous studies have shown that focal muscle vibration (fMV) is able to modulate inhibitory interneuron activity of the primary motor cortex (M1), thus increasing M1 excitability [14,19]. The authors suggested that changes in sensory inputs from muscle spindles induced by fMV remodel the interaction between sensory inputs and motor output [14,19], possibly shaping the excitability of inhibitory GABAergic circuits in M1 [14,19]. In addition, it has been demonstrated that these changes in M1 are specific to the vibrated muscle, which may reflect changes in sensory representations induced by fMV at the cortical and subcortical levels [14,19,20]. The same specific effect was also described in a recent study, showing that fMV was able to induce plasticity in spinal cord circuits [21].

Based on this evidence, it is reasonable to hypothesize that if fMV can modulate sensory afferents, motor outputs and their integration, then it may also induce changes in mechanisms of sensorimotor integration such as those involved in the STDT–movement integration task.

Therefore, the aim of the present study was to assess whether fMV, applied over the first dorsalis interosseous (FDI) muscle, would modulate STDT values tested during and after voluntary movement execution in healthy subjects. It has been also tested whether fMV applied on FDI can induce changes in the STDT–movement integration. In order to better clarify whether the effect of fMV on STDT–movement integration depended on the vibrated muscle, in a control experiment, fMV was applied over the abductor pollicis brevis (APB) or the flexor radialis carpi (FRC) muscles. Finally, in order to investigate the effect of the amplitude of the vibratory stimulus on the STDT–movement integration changes, fMV was used to generate a different displacement in a separate control experiment. These data may clarify the mechanism of action of the fMV in physiological conditions, and whether fMV might have therapeutic applications in movement disorders that are known to underlie abnormal sensorimotor integration.

In the following sections, the methodological approach used in the study, including details on the experimental procedure, will first be described; afterwards, results will be reported, explained and discussed in the following “results” and “discussion” sessions.

## 2. Materials and Methods

The effects of FMV on STDT and STDT–movement integration were investigated in a sample of healthy volunteers. To follow, demographic features of the volunteers, inclusion and exclusion criteria and methodological details for each experimental procedure will be detailed.

### 2.1. Participants

A total of 17 healthy volunteers were enrolled in the study (9 females, 8 males; mean sample age: 28.83 ± 1.62 years).

The sample size was based on an a priori analysis with G*Power 3.1 software, assuming an expected effect size (Cohen’s d) of 0.85 and a statistical power of 0.80 at a 0.05 alpha level, with the number of measurement set as 4 for experiment 1 and as 3 for experiments 2 and 3. This resulted in a total sample size of 17 subjects for experiment 1 and 15 subjects for experiments 2 and 3.

Accordingly, all participants took part in experiment 1, while only fifteen of them took part in experiments 2 and 3.

Exclusion criteria were a history of neuropsychiatric disorders or symptoms of focal upper limb nerve entrapment, cervical brachialgia, polyneuropathy or central nervous system (CNS)-acting drug intake at the time of the experiments. Volunteers were asked to avoid the intake of caffeine, nicotine or any other substances interfering with CNS at least 6 h before the experimental session.

All participants except two were right-handed, as evaluated by the Edinburgh Handedness Inventory [22]. All experimental procedures were performed in accordance with the Declaration of Helsinki. The study was approved by the ethics committee of Sapienza, University of Rome, and informed consent was given by all study participants. All volunteers were naive to fMV and STDT protocols.

### 2.2. Electromyography (EMG)

EMG activity was recorded through a pair of surface electrodes placed over the FDI muscle of the dominant hand, in a belly–tendon configuration. The EMG signal was recorded (×1000) and filtered (5–5000 Hz) with a Digitimer D360 (Digitimer Ltd., Hertfordshire, UK; bandwidth 20 Hz–1 kHz), and then analyzed offline with a personal computer through a 1401 plus analog–digital (A/D) laboratory interface (Cambridge Electronic Design, Cambridge, UK) [3]. Data were stored on a laboratory computer for online visual display and further offline analysis (Signal software; Cambridge Electronic Design). EMG activity from the FDI muscles during voluntary movement was measured by assessing the root mean square amplitude (see *STDT–movement integration paradigm* paragraph).

### 2.3. STDT

STDT was investigated according to previous studies [2,3,4], using paired tactile stimuli consisting of square-wave electrical pulses delivered using a constant current stimulator (Digitimer DS7A; Digitimer Ltd.) through surface electrodes placed over the distal phalanx of the index finger of the dominant hand. Stimulation intensity was defined for each participant by delivering a series of stimuli, starting at an intensity of 2.0 mA and progressively increasing the intensity in 0.5 mA steps. The intensity used for STDT was set at the minimal intensity the participant perceived in 10 of 10 consecutive stimuli. Paired stimuli were delivered starting from an interstimulus interval (ISI) of 0 milliseconds (ms) and progressively increasing the ISI in 10-millisecond steps. The first of three consecutive ISIs at which participants recognized the stimuli as temporally separate was considered the STDT. The STDT value was defined as the average of three STDT trials, and was then entered into the data analysis.

### 2.4. STDT–Movement Integration Paradigm

Similarly to previous studies [7,9], paired stimuli for STDT testing were triggered by movement execution at EMG onset in the FDI and at three intervals thereafter, as soon as the EMG signals reached 100 uV in amplitude (defined as “0 ms”) and at 100 ms and 200 ms afterwards. Participants were instructed to abduct the index finger after a verbal go signal, and they were asked to perform the movement as largely and as quickly as possible. STDT values were assessed on the volar surface of the right index finger at movement onset (0 ms) and at 100 ms and 200 ms afterwards. The STDT value for each of the ISIs was defined as the average of three trials.

### 2.5. FMV

Focal muscle vibration was applied to the FDI muscle using a specific device consisting of an electromechanical transducer, a mechanical support and an electronic control (CroSystem, NEMOCO).

The instrument consisted of an electromechanical transducer, a specific mechanical support and an electronic control device. The transducer was positioned over the FDI muscle at a point approximately corresponding to its maximal transversal size. The FDI was chosen as it is feasible for the fMV protocol and it is involved in the motor task used for the STDT–movement integration paradigm.

The vibration was applied by means of a small probe 10 mm in diameter directly over the FDI muscle belly. The transducer was driven to produce sinusoidally modulated forces. Soft tissues were compressed to ensure better transmission of the vibrations into the muscle [23,24,25].

The vibration frequency was set at 100 Hz and vibration amplitude ranged from 0.1 to 0.2 mm peak-to-peak, which was sufficient to drive Ia spindle afferents, to avoid muscle fiber injury and as a subthreshold for the tonic vibration reflex. Each instance of stimulation consisted of two blocks lasting 10 min (‘), with an interblock interval of 1 min.

### 2.6. Experimental Paradigm

The design of the study included a main experiment (experiment 1) and two control experiments (experiments 2 and 3), which took place one week apart from the main experiment to avoid interference due to possible long term effects of each fMV session. The order of the experimental sessions was randomized across the participants.


*Experiment 1: Effects of fMV applied over the FDI muscle on STDT and STDT–movement integration*


All subjects participated at this experiment. STDT and the STDT–movement paradigm were tested prior to, during and after the application of fMV over the FDI muscle, which was involved in the motor task underlying STDT–movement integration. In particular, the STDT at rest and the STDT–movement paradigm were tested at the beginning of each experimental session (T0), during the second block of fMV (T1) and 5′ (T2) and 10′ (T3) after fMV. FMV was applied to generate a 0.1 mm peak-to-peak sinusoidal displacement. For the STDT–movement paradigm, the different intervals (0, 100 and 200 ms) between movement onset and the STDT testing were selected in a randomized order for each subject (Figure 1).


*Experiment 2 (fMV topographic specificity): effects of fMV applied over the abductor pollicis brevis (APB) or the flexor radialis carpi (FRC) muscles on STDT and STDT–movement integration*


In order to investigate whether the possible effects of fMV on STDT or STDT–movement changes were specific to the stimulated muscular district, STDT and STDT–movement integration were tested prior to, during and after the application of fMV on different muscles of the upper limbs that were not involved in the movement execution. To this aim, in 14 out of 17 participants enrolled in experiment 1, the STDT and the STDT–movement paradigm were tested “before”, “during” and “after” fMV was applied over the APB, as well as over the FRC, in separate experimental sessions. The stimulation parameters and time points were the same as those used for experiment 1.


*Experiment 3: Effects of fMV applied over the APB generating a 0.2 mm muscle displacement*


In order to investigate whether possible effects of fMV on STDT or STDT–movement integration depended on vibration parameters, STDT and STDT–movement integration were tested prior to, during and after the application of fMV, which generated a 0.2 mm displacement of the APB muscle. To this aim, in 14 out of 17 participants enrolled in experiment 1, the STDT and the STDT–movement paradigm were tested “before”, “during” and “after” fMV was applied, generating a 0.2 mm displacement of the APB muscle. The stimulation parameters and time points were the same as those used for experiment 1.

### 2.7. Statistical Analysis

Statistical analysis was performed using SPSS 20 software (SPSS Inc, Chicago, IL, USA). Repeated measures (RM) analysis of variance (ANOVA) and post hoc analyses were conducted by means of *t*-tests with Bonferroni’s correction for multiple comparisons. Compound symmetry was calculated with Mauchly’s test, and the Greenhouse–Geisser correction was applied when required. All variables were first tested for normality using the Kolmogorov–Smirnov test. Significance was set at *p* value < 0.05. Unless otherwise stated, values are expressed as means ± standard error of the mean (SEM). To quantify the magnitude of effect for the observed differences, the partial eta-squared effect size (ηp2) was computed.

For all the experiments, to evaluate changes in STDT value due to fMV, a one-way RM-ANOVA was used, with time (before (T0), during (T1) and 5′ (T2) and 10′ (T3) after fMV) as a within-subject factor. For the STDT–movement integration, to evaluate changes in STDT values during movement execution as determined by fMV, a two-way RM-ANOVA was performed with time (before (T0), during (T1) and 5′ (T2) and 10′ after (T3) fMV for experiment 1; and only T0, T1 and T2 for experiments 2 and 3) and ISI (baseline, 0 ms, 100 ms and 200 ms) as within-subject factors.

## 3. Results

In the following section, the results obtained from each experimental session will be described. All the results are summarized in Table 1.


*Experiment 1: Effects of fMV applied over the FDI muscle on STDT and STDT–movement integration*


The one-way RM-ANOVA performed to investigate changes in STDT values while fMV was applied over the FDI failed to detect a significant effect of the time factor (F(_3,39)_ = 1.239; *p* = 0.299, ηp2 = 0.072), showing no differences in STDT values measured at rest before, during or after fMV (Figure 2). The two-way RM-ANOVA, performed to compare STDT–movement integration before, during and after fMV was applied over the FDI, showed significant effects of ISI (F_(3,39)_ = 57.739; *p* < 0.001, ηp2 = 0.783), time (F_(3,39)_ = 34.894; *p* < 0.001, ηp2 = 0.686) and interaction between factors (F_(9,117)_ = 8.69; *p* < 0.001, ηp2 = 0.352) (Figure 3). First, the post hoc analysis showed a significant increase in STDT values at different ISIs from movement onset compared to at rest, showing significant changes in STDT values during movement as well as before (rest vs. 0 ms: *p* < 0.001; rest vs. 100 ms: *p* < 0.001), during (rest vs. 0 ms: *p* < 0.001, rest vs. 100 ms: *p* < 0.001 and rest vs. 200 ms: *p* = 0.01) and after (rest vs. 0 ms: *p* < 0.001, rest vs. 100 ms: *p* < 0.001 and rest vs. 200 ms: *p* = 0.04) fMV. Moreover, the post hoc analysis showed that STDT–movement integration at T1 (during fMV) significantly differed from that at T0 (before fMV) (*p* = 0.001). Similarly, STDT–movement at T2 (5 ‘ after fMV) (*p* < 0.001), as well as at T3 (10′ after fMV), was significantly different from that at T1 (during fMV) (*p* < 0.001). Finally, the post hoc analysis showed a significant effect of fMV on STDT modulation only “during” fMV at all the ISIs (0 ms, 100 ms and 200 ms) after index finger abduction (all *p* < 0.05).


*Experiment 2: fMV topographic specificity effects of fMV applied over the APB or the FRC muscles on STDT–movement integration task*


The one-way RM-ANOVA performed to investigate changes in STDT values at rest when fMV was applied over the FDI muscle showed a non-significant effect of time (F_(3,39)_ = 2.199; *p* = 0.139, ηp2 = 0.136). The two-way RM-ANOVA performed to compare STDT–movement integration before, during and after fMV’s application over the FDI showed a significant effect of ISI (F_(3,39)_ = 23.517; *p* < 0.001, ηp2 = 0.681), but a non-significant effect of time (F_(3,39)_ = 1.885; *p* = 0.179, ηp2 = 0.146) and of interaction between factors (F_(3,39)_ = 0.917; *p* = 0.459, ηp2 = 0.077). Post hoc analysis showed a significant difference between the baseline and 0 ms, 100 and 200 ms (*p* < 0.05). Similarly, in the experimental session in which fMV was applied over the FRC muscle, the one-way RM-ANOVA showed a non-significant effect of time (F_(3,39)_ = 0.043; *p* = 0.956; ηp2 = 0.003) for STDT values. In addition, the two-way RM-ANOVA on STDT–movement integration showed a significant effect of ISI (F_(3,39)_ = 30.851; *p* < 0.001, ηp2 = 0.720), but a non-significant effect of time (F_(3,39)_ = 0.811; *p* = 0.411, ηp2 = 0.063) and of interaction between factors (F_(3,39)_ = 0.517; *p* = 0.738, ηp2 = 0.041). Post hoc analysis still showed a difference between baseline, 0 ms and 100 ms (*p* < 0.001 and *p* = 0.02, respectively).


*Experiment 3: Effects of fMV applied over the APB generating a different muscle displacement*


The one-way RM-ANOVA performed to investigate changes in STDT values at rest during the application of fMV, generating a 0.2 mm displacement of the APB muscle, showed a non-significant effect of time (F_(3,39)_ = 2.199; *p* = 0.139, ηp2 = 0.136). The two-way RM-ANOVA performed to compare STDT–movement integration before, during and after the application of fMV, generating a 0.2 mm displacement of the APB muscle, showed a significant effect of ISI (F_(3,39)_ = 23.517; *p* < 0.001, ηp2 = 0.681), but a non-significant effect of time (F_(3,39)_ = 1.885; *p* = 0.179, ηp2 = 146) and of interaction between factors (F_(3,39)_ = 0.917; *p* = 0.459, ηp2 = 0.077). Post hoc analysis showed significant differences between the baseline, 0 ms, 100 ms and 200 ms (*p* < 0.05).

## 4. Discussion

The present study demonstrates that fMV does not influence STDT at rest, but significantly modulates STDT–motor integration. FMV-induced modulation was observed only during fMV administration, and was specific to the muscle involved in the motor task performed to evaluate STDT–motor integration.

To rule out the possibility that movement-induced STDT modulation evaluated before, during and after fMV was due to changes in attention, the ISIs for the STDT–movement protocol were tested in a randomized order, except for the STDT measurement at rest, which was performed as the first assessment in all trials. In addition, the order of the experimental sessions was randomized across the participants. The fMV protocol used in our study was selected in order to obtain the highest muscle–tendon complex displacement without evoking reflex muscular contractions, i.e., the tonic vibration reflex, or selectively stimulating the Ia afferents [24]. As the fMV-induced changes in STDT–movement integration were observed only when fMV was applied on the FDI and not on the APB or FCR muscles, it may be ruled out that fMV-induced effects reflect non-specific interference with the sensorimotor task.

The observation that fMV modified STDT in healthy subjects only when tested during movement, but not at rest, suggests that the fMV is able to interfere with mechanisms of sensorimotor integration, but not with pure sensory processing in the temporal domain. The modulation of STDT–movement integration observed during fMV may possibly reflect an interaction between the neural circuits involved in the STDT–movement integration task and the proprioceptive circuits activated by fMV in physiological conditions.

One possible explanation for our findings is that vibratory stimulation may affect cortical processing of sensorimotor information related to movement execution. However, the observation that fMV left STDT values unchanged tentatively rules out that changes in STDT–movement integration during fMV rely on inhibitory circuits in S1 [4,5]. Moreover, data showing that muscle vibration produces Ia afferent input that reaches the areas 3a and 4 of S1 [26], whereas the input from cutaneous receptors primarily reaches the areas 3b and 1 of S1 [26,27], support the hypothesis that interplay between fMV and STDT–movement occurs at a CNS level other than S1.

Another possible explanation is that fMV interferes with mechanisms of STDT–movement integration by altering M1 excitability. According to previous studies, fMV selectively increases the excitability of the representation of the vibrated muscle in M1 [20]. Our observation that fMV only modulated STDT–movement integration when vibration was applied over the muscle involved in the movement, but not when applied over other muscles, might lead us to hypothesize that fMV interferes with cortical mechanisms. It is, however, unclear whether the topographic specificity of fMV-induced effects relies on cortical mechanisms or indirectly depends on subcortical processing. FMV may indeed act by interfering with the physiological sensory gating that takes place during the STDT–movement integration. In healthy humans, during movement execution, information related to the incoming movement is projected to sensory areas, including basal ganglia–thalamus circuits [28] and S1, in order to tailor the motor command to sensory expectations. Accordingly, previous studies have shown that during movement execution, STDT-related tactile information is gated to prioritize movement-related sensory inflow [7,9]. Previously, several works have suggested that this process takes place at the subcortical level, and specifically in striatum–thalamus circuits [7,8].

Studies on monkeys showed that the putamen and globus pallidus process Ia afferent inputs [29,30]. Moreover, neuroimaging studies in healthy subjects which were performed using fMRI also showed the activation of the putamen, pallidum and thalamus [28,31] during muscle spindle stimulation. Hence, it can be suggested that vibratory muscle stimulation interferes with STDT–movement task processing, modulating the physiological gating of sensory information at the basal ganglia–thalamic level [7]. The vibration-related proprioceptive inflow from the body region involved in the movement may be prioritized at the expense of the STDT-related tactile afferents, thus impairing the STDT–movement integration process. According to this hypothesis, the modulation of STDT values induced by fMV was temporally locked to movement onset, with the STDT returning to baseline values when tested 200 ms after movement onset. In addition, the fMV-induced changes in STDT–movement integration only occurred when vibration was applied over the muscle involved in voluntary movement execution.

The muscle spindle input activation determined by fMV, as well as subsequent changes in sensory gating at the subcortical level, could also potentially have an influence on the sensorimotor integration processes underlying STDT–movement integration at the cortical level [17]. FMV-induced proprioceptive inputs and STDT-related tactile inputs may converge on a selected population of cortical interneurons specifically responsible for sensorimotor integration [14,19].

Finally, a further relevant observation of this study was that the effects observed in the present study on STDT–movement integration only lasted for the duration of the stimulation, without inducing plasticity. This is apparently in contrast with a previous study which demonstrated that fMV can induce long-term depression-like plasticity in healthy subjects [32]. However, this study showed that fMV determines changes in neurophysiological parameters that are encoded at the spinal level [32]. Contrastingly, sensory processing in the temporal domain [33], and specifically STDT–movement integration [9], are encoded at the basal ganglia–thalamic level. Thus, it is possible that the lack of long-term effects observed in our study was due to the neurophysiological circuit which was engaged.

### Limitations 

We acknowledge some limitations of this study. First, we did not measure kinematic features of movement, which would have clarified whether fMV induced changes in motor performance. A further limitation is that we did not test M1 excitability [33]; thus, considerations of fMV’s effects on M1 can only be speculative. Finally, this study did not analyze the possible effects of fMV on the EMG signal, which can be practically performed by future studies [34].

## 5. Conclusions

Nevertheless, our results have potential implication for future studies aiming to investigate the effects of fMV in patients with dystonia, a movement disorder characterized by altered sensorimotor integration and gating of sensory processing [9,35]. In conclusion, this study demonstrated that FMV is able to modulate sensorimotor integration, as measured by the STDT–movement task, when applied over the muscle involved in the motor task. This result has potential implications for future studies aiming to investigate the effects of fMV in patients with dystonia, a movement disorder characterized by altered sensorimotor integration and gating of sensory processing [9,35].

## Figures and Tables

**Figure 1 brainsci-13-00664-f001:**
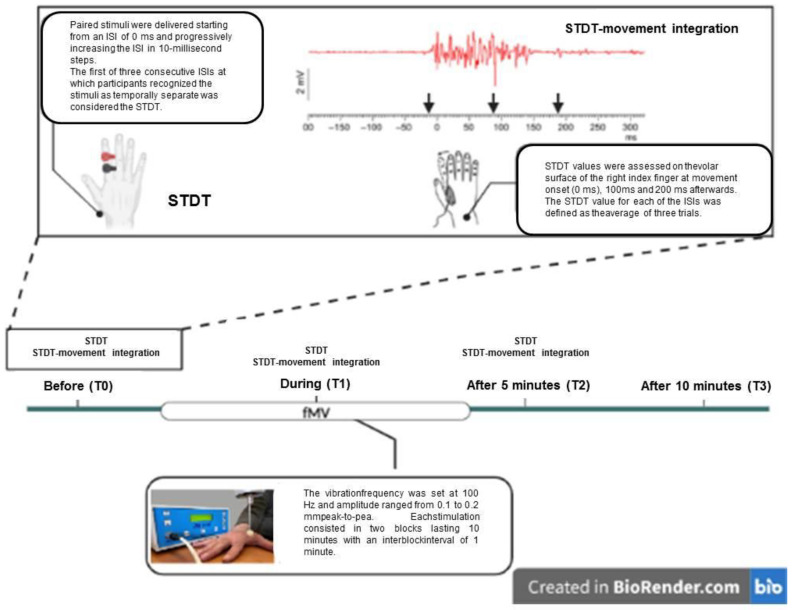
Experimental procedure. Created with https://www.biorender.com/, accessed on 28 March 2023.

**Figure 2 brainsci-13-00664-f002:**
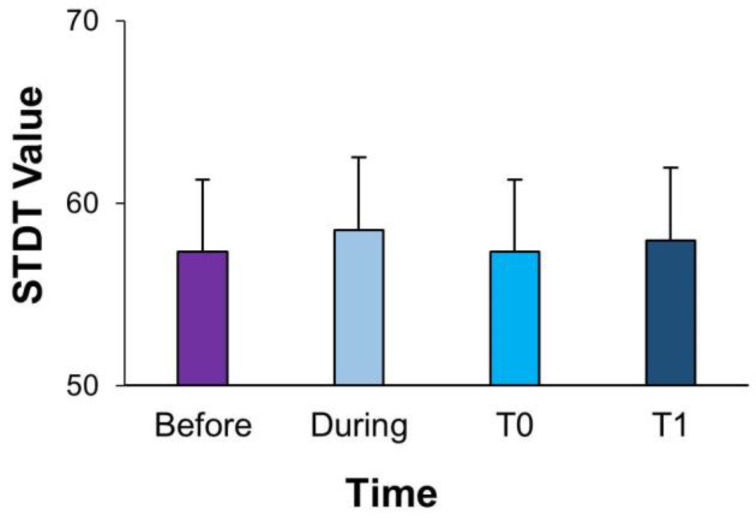
STDT values measured at rest at four different time points: before (T0), during (T1) and 5′ (T2) and 10′ (T3) after focal muscle vibration (fMV). FMV did not induce significant changes in STDT values at rest.

**Figure 3 brainsci-13-00664-f003:**
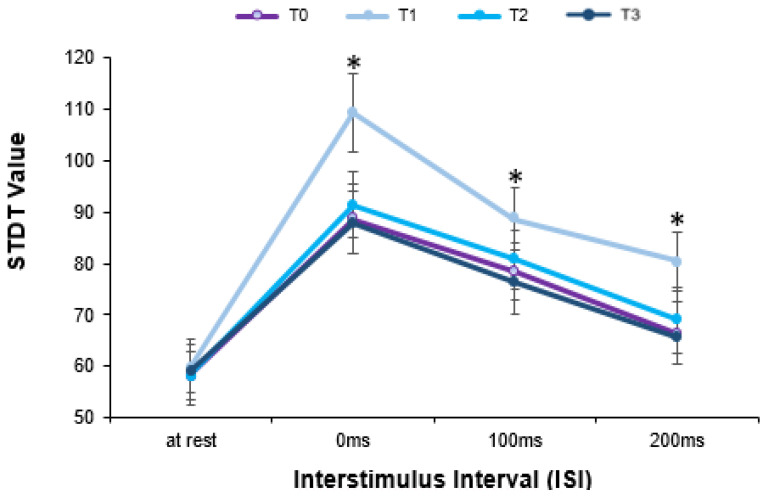
The effect of focal muscle vibration (fMV) applied over the FDI on somatosensory temporal discrimination threshold (STDT)–movement integration. STDT–movement integration was tested as movement-induced changes in STDT values at movement onset (0 milliseconds) and at 100 milliseconds and 200 milliseconds afterward at different time points: before (T0), during (T1) and 5 min (T2) and 10 min (T3) after fMV. Movement-induced STDT modulation during FDI-fMV (line in cyan) significantly differed from that obtained before (line in purple), 5′ after (line in light blue) and 10′ after fMV (line in dark blue). Asterisks (*) highlights significant results.

**Table 1 brainsci-13-00664-t001:** STDT parameters.

Protocol		Experiment 1				Experiment 2						Experiment 3		
	Muscle fMV	First Dorsali Interosseus				Abductor Pollicis Brevis			Flexor Radialis Carpi			Abductor Pollicis Brevis		
	Time	T0	T1	T2	T3	T0	T1	T2	T0	T1	T2	T0	T1	T2
STDT-At rest		57.35 ± 3.93	58.53 ± 3.97	57.35 ± 3.93	57.94 ± 4.0	49.77 ± 3.23	52.44 ± 2.75	52.00 ± 2.96	52.33 ± 2.80	53.00 ± 3.44	52.67 ± 3.30	49.77 ± 3.23	52.44 ± 2.75	52.00 ± 2.96
**STDT-movement integration**														
0 ms		87.65 ± 5.79	107.06 ± 6.57	90.59 ± 5.52	85.88 ± 5.56	88.33 ± 5.91	88.00 ± 6.38	86.25 ± 6.59	83.67 ± 5.08	84.00 ± 4.34	83.46 ± 4.84	88.33 ± 5.91	88.00 ± 6.38	86.25 ± 6.59
100 ms		78.24 ± 4.72	89.41 ± 5.11	80.00 ± 5.0	77.65 ± 5.32	75.44 ± 4.43	78.07 ± 3.56	76.11 ± 3.62	76.0 ± 5.24	78.0 ± 5.21	72.68 ± 4.40	75.44 ± 4.43	78.07 ± 3.56	76.11 ± 3.62
200 ms		65.29 ± 5.21	80.29 ± 4.90	67.35 ± 5.46	64.71 ± 5.41	61.00 ± 4.12	65.33 ± 4.84	64.17 ± 4.50	61.67 ± 5.06	64.00 ± 5.76	60.38 ± 5.12	61.00 ± 4.12	65.33 ± 4.84	64.17 ± 4.50

STDT, Somatosensory temporal discrimination threshold; ms, milliseconds. The table reports means ± standard mean error (SEM).

## Data Availability

Individual data is unavailable due to privacy restrictions. Aggregated data are available from the corresponding author upon reasonable request.

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
