# Peer review of "Investigating the Effects of a Focal Muscle Vibration Protocol on Sensorimotor Integration in Healthy Subjects"

_brainsci, 2023, doi:10.3390/brainsci13040664_

Round 1

Reviewer 1 Report

This is a very interesting study that aimed to investigate the effects of a focal muscle vibration protocol in healthy participants. The study is relevant, original and offers a contribution to the knowledge of the area. In that sense, I congratulate the authors. The manuscript, however, needs to be extensively revised and some suggestions and questions are presented here for good application of findings.

 1. Title:

1.1. Clarify the target population.

 2. Abstract:

2.1. Avoid abbreviations. There are too many abbreviations in the abstract section.

2.2. I am not sure if the results allow the statement “….disrupting the physiological gating of sensory information at basal ganglia-thalamic level”. Please rewrite this sentence.

2.3. Conclusion is not answering the aim of the study.

 3. Introduction:

3.1. Please explain why you recruited healthy participants, if previous studies reported abnormal STDT-movement integration in basal ganglia disorders.

3.2. Furthermore, explain why the use of fist dorsalis interosseous muscle, instead of other muscles.

 4. Methods:

4.1. Please include a sample size calculation. Fourteen participants seem fragile in a sample of healthy subjects.

4.2. Include inclusion criteria.

4.3. Insert flow diagram of the recruitment.

4.4. Insert figures that could clarify the methodological procedures.

4.5. Please explain why the use of fist dorsalis interosseous muscle instead of other muscles.

4.6. Did you perform normality tests? Data from 14 subjects is more likely to be non-parametric.

 5. Results:

5.1. Is it possible to provide effect sizes and statistical power of the RM-ANOVA analyses?

5.2. Figures should be included after the text and not in a different topic.

 6. Discussion: Adequate.

 7. Conclusion:

7.1. Is there not a conclusion section?

 8. Institution Review Board.

8.1. Explain why the study has been submitted now, if the Institution Review Board approved the study 10 years ago (2013)?

 9. References

9.1. Avoid self-citation. There are 7 studies from a total of 32 of the research group. It is not polite to do that. Is the same ethics protocol approved in 2013 used in the 7 studies?

Reviewer 2 Report

The authors presents the article entitled “Investigating the effects of a focal muscle vibration protocol on sensorimotor integration”

This paper assesses whether the fMV applied over the first dorsalis interosseous (FDI) muscle, modulates STDT values tested during and after voluntary movement execution. Reviews whether any possible fMV-induced changes on the STDT-movement integration depend on the vibrated muscle, and the amplitude of the vibratory stimulus. These data may clarify whether fMV might have therapeutic applications in movement disorders that are known to underlie an abnormal sensorimotor integration.

The article presents the following concerns:

  • The text must be written in the third person or passive voice.
  • It is recommended at the end of the introduction to describe the structure of the text, indicating the content of each point.
  • Literature should be updated using up-to-date references. 50% of the references are more than ten years old.
  • It is recommended to add a short introduction between section 2 and subsection 2.1
  • Authors are requested to add a comparative table between the results and achievements of this work concerning related articles.
  • Add the age range of the male participants or clarify if the ages mentioned apply to all participants.
  • It is essential to justify the reason for the proposed bandwidth, such as the sampling frequency used and the measurement duration.
  • Add a short introduction between section 3 and subsection 3.1
  • Authors are encouraged to organize results in table format.
  • According to Turnitin, the similarity value should be reduced from 44% to less than 20%.
  • The authors are invited to detail further the methodology implemented for the STDT, such as the conditions in which the volunteers arrived, if any medication or food was excluded before the tests, etc.
  • EMG background can be threatened before line 91 to inform the readers of the theory behind these signals. It can be discussed with the following references: A novel methodology for classifying emg movements based on svm and genetic algorithms; Support vector machine-based emg signal classification techniques: a review.
  • Section 2.4: The discussion or conclusions section should mention the disadvantages and possible errors the time between when the verbal instruction is given, and the volunteer performing the movement brings with it. What happens if the volunteer takes more than the defined time? How long is it valid for the movement to be carried out after the order? etc
  • Section 2.5: It is crucial to define and mention the methodology followed for the placement of the transducer on the muscle, the importance should be discussed, and what differences the results present depending on the variation in the position in its placement.
  • Why was a space of one week selected between stimuli? What consequences would it have to carry them out early or late? It is essential to justify the methodology and rely on previous work so that the authors can add references in section 2.6.
  • Authors are asked to analyze and describe figure 2, which is only mentioned at the end of line 207.

The following misspelling should be checked:

  1. line 64: The adverbial “also” appears to be misplaced in this sentence. Determine the appropriate placement for the adverb. “also described”
  2. lines 119-120: Your sentence may be unclear or hard to follow. Consider rephrasing. “After a verbal go signal, participants were instructed to abduct the index finger as wider and faster as possible”

Round 2

Reviewer 1 Report

The authors did a great job in improving the quality of the manuscript. I am satisfied with the study in its current form.

Reviewer 2 Report

The manuscript can be accepted